# Dietary Alpha-Linolenic Acid Supports High Retinal DHA Levels

**DOI:** 10.3390/nu14020301

**Published:** 2022-01-12

**Authors:** Andrew J. Sinclair, Xiao-Fei Guo, Lavinia Abedin

**Affiliations:** 1Department of Nutrition, Dietetics and Food, School of Clinical Sciences, Monash University, Melbourne, VIC 3168, Australia; 2Institute of Nutrition & Health, College of Public Health, Qingdao University, Qingdao 266071, China; guoxf@qdu.edu.cn; 3Department of Food Science and Technology, School of Science, RMIT University, Melbourne, VIC 3001, Australia; laviniaabedin@gmail.com

**Keywords:** omega-3, docosahexaenoic acid, alpha-linolenic acid, retina, brain, liver, diet, linoleic acid, linoleic acid to linolenic acid ratio, guinea pigs, equivalence, biomarker, docosapentaenoic acid n-6, 22:5n-6, vegans, vegetarians

## Abstract

The retina requires docosahexaenoic acid (DHA) for optimal function. Alpha-linolenic acid (ALA) and DHA are dietary sources of retinal DHA. This research investigated optimizing retinal DHA using dietary ALA. Previous research identified 19% DHA in retinal phospholipids was associated with optimal retinal function in guinea pigs. Pregnant guinea pigs were fed dietary ALA from 2.8% to 17.3% of diet fatty acids, at a constant level of linoleic acid (LA) of 18% for the last one third of gestation and retinal DHA levels were assessed in 3-week-old offspring maintained on the same diets as their mothers. Retinal DHA increased in a linear fashion with the maximum on the diet with LA:ALA of 1:1. Feeding diets with LA:ALA of 1:1 during pregnancy and assessing retinal DHA in 3-week-old offspring was associated with optimized retinal DHA levels. We speculate that the current intakes of ALA in human diets, especially in relation to LA intakes, are inadequate to support high DHA levels in the retina.

## 1. Introduction

Docosahexaenoic acid (DHA, 22:6n-3) is concentrated in the disk membranes of rod outer segments of photoreceptor cells in the retina [1]. The high proportions of DHA in these membranes allows efficient conformational changes to occur in rhodopsin (the light receptor) during phototransduction [2]. Wheeler et al. [3] showed that the electrical response of the photoreceptor cell was a relative function of dietary alpha-linolenic acid (ALA) and linoleic acid (LA) content, with the greatest response being to diets containing ALA. These data showed a selective functional role for omega 3 PUFA, presumably because of the very high DHA content of retina membrane phospholipids (PL).

Both the brain and retina are enriched in DHA compared with other tissues [4] and have a limited capacity for DHA synthesis [5]. The retina and brain acquire DHA either from DHA synthesized endogenously by desaturation and chain elongation from ALA in the liver (Appendix A, Figure A1) or by consumption of DHA from diet. In the rat, it has been reported that the liver capacity to synthesize DHA (by the desaturation—chain elongation pathway) far exceeds the requirements of the brain for DHA (5). This is presumably also true for retinal DHA.

Prolonged dietary deficiency of ALA results in substantial losses of DHA from the retinal membranes and this is associated with sub-optimal responses of the retina (rhodopsin) to light. This has been reported in a variety of animal species, including primates, rats, cats and guinea pigs [6]. In all these studies, the ALA deficient diets were rich in LA, which in the absence of dietary ALA, is converted to the LA metabolite, 22:5n-6, synthesized by the same desaturation—chain elongation pathway which forms DHA from ALA (Figure A1). Therefore, tissue levels of 22:5n-6 are widely regarded as a biochemical indicator of a deficiency of ALA (and DHA) in the diet [7].

Typically, dietary deficiency studies have reported depletions of retinal DHA of more than 85%, which is almost reciprocally replaced by 22:5n-6. For example, in the guinea pig, we have shown that at 6–9 weeks-of-age on an ALA-deficient diet (3rd generation), the retinal DHA was 2.5% compared with 21% in control animals, while the 22:5n-6 levels were 23% (deficient) and 5% (control) [8]. In vitro studies have established that 22:5n-6 is not an effective replacement for DHA in terms of optimizing the function of rhodopsin [9].

In studies of ALA deficiency in animals, no overt symptoms of the deficiency have been described [7], unlike essential fatty acid deficiency where animals show reduced weight, scaly skin, dandruff and loss of fertility. To study ALA deficiency, animals are fed diets containing vegetable oils extremely rich in LA and almost devoid of ALA, such as safflower oil, peanut oil or corn oil; the control diets used ALA-rich oils such as canola oil, rapeseed oil or soybean oil or fish oil (containing DHA); thus, the visual function studies have compared two levels of retinal DHA—very low in the case of the ALA-deficient group or high for the control group. One dose–response study of retinal DHA levels versus visual function in guinea pigs [10] measured visual function in animals which had retinal DHA values ranging from 2.5% (ALA-deficient group) to 30.8% (fish oil group); two other groups had retinal DHA values of 21.0% (canola oil group) and 12.4% (Lab Chow group). Trend analysis modelling showed that retinal function was altered by tissue DHA as described by an inverted “U-shaped” function, with the optimal function occurring when retinal DHA was 19% of PL fatty acids [10].

What is not known is the minimum amount of dietary ALA to reach a retinal DHA value within the optimal retinal function range (19% DHA). We have shown in guinea pigs that ALA at 1% diet fatty acids (in a diet with LA:ALA of 17.3) was not sufficient to achieve the optimum retinal DHA levels (retinal DHA reached 9.6%), while a diet containing ALA at 7.1% (LA:ALA = 2.7) reached the target range of retinal DHA level (retinal DHA reached 16.4%) [11].

Therefore, the aim of this study was to conduct an ALA-dose–response study to assess the level of dietary ALA necessary to increase retinal DHA to the optimal retinal DHA level and to minimize retinal 22:5n-6 in guinea pigs. To achieve this, pregnant guinea pigs were fed diets with increasing amounts of ALA at a constant LA level and the retina PL DHA and 22:5n-6 levels were measured in 3-week-old offspring. The hypothesis was that high ALA diets in a ratio with dietary LA of approximately 2:1 would lead to optimal retinal DHA levels.

While there are similarities between humans and guinea pigs in terms of brain development in utero [12], using the timing for peaks in brain growth velocity as a marker of development, guinea pigs can be classified as prenatal brain developers, while humans are considered perinatal brain developers (and other rodents as postnatal brain developers) [13]. As with humans, the rapid phase of myelination and synapse formation is initiated in the last half of pregnancy in guinea pigs [12].

## 2. Materials and Methods

### 2.1. Animals and Diets

The study was conducted on Duncan Hartley guinea pigs and was approved by RMIT University Animal Ethics Committee.

In this study, 21 guinea pigs (16 female and 5 male) were randomly divided into 5 groups consisting of 3–4 females per male guinea pig per group (4 groups of 3 females and 1 male; 1 group of 4 females and 1 male) and fed commercial chow diets. Once pregnancy was confirmed by the presence of a vaginal plug, the females were maintained on the commercial chow diets for the first 2/3 of pregnancy (until embryo day 46) and then the females in the 5 groups were placed on one of five semi-synthetic diets after that time until 3 weeks after delivery. The offspring were housed with the dams in their respective diet groups until the offspring were 3-weeks of age. The gestation period in the guinea pig is typically 69–71 days [12]. This relatively long gestation and the maturity of the guinea pig pups at birth means that a significant proportion of brain development, including myelination, happens in utero [14].

The animals were supplemented with fresh carrots and drinking water containing ascorbic acid (400 mg/L). All diets contained 10% lipid (*w*/*w*); the 1st diet used safflower oil as its source of lipids and was used to provide a reference point for an ALA-deficient diet, and in the other 4 diets the lipids were supplied by a mixture of vegetable oils (canola, coconut, palm stearine, safflower, sunola and linseed oils) to maintain a constant LA proportion (18% FA, range 17.5–18.6%) with varying ALA content (0.7% to 17.3%) (Appendix A, Table A1). The macro- and micronutrient content of the diets was as described by Weisinger et al. [8]. Three weeks after birth, offspring were sacrificed using carbon dioxide asphyxiation and the livers and brains were removed, washed in saline, blotted dry and maintained at −70 °C for later fatty acid analysis. The retinas were quickly removed, washed in ice-cold phosphate-buffered saline and stored in chloroform/methanol (2:1 *v*/*v* containing 10 mg/L of butylated hydroxytoluene), at −70 °C for later fatty acid analysis.

### 2.2. Analyses

Following lipid extraction of 1 g of liver and brain and the retinas in chloroform-methanol (2:1 *v*/*v* containing 10 mg/L of butylated hydroxytoluene), the PL were separated from neutral lipids by TLC and the fatty acid methyl esters of the PL fraction were prepared and separated by gas chromatography with FID on a BPX-70 capillary column [8]. FAME were identified by comparison with fatty acid methyl ester standard mixtures and results calculated by use of response factors derived from chromatographing fatty acid methyl ester standard mixtures of known composition (Nu-Chek Prep. Inc., Elysian, MN, USA).

### 2.3. Statistical Analyses

Significant differences between dietary groups were tested using a one-way analysis of variance. Post-hoc comparisons were made using the Tukey test with a significance level of 0.05.

## 3. Results

### 3.1. Diets

The ALA proportions in the five diets ranged from 0.7% to 17.3%, with LA to ALA ratios from 103:1 to 1.05:1 (Appendix A, Table A1).

### 3.2. Effect of Increasing Dietary ALA on Tissue Fatty Acids in 3-Week-Old Guinea Pigs

This experiment involved feeding five diets with increasing proportions of ALA to pregnant guinea pigs and assessing tissue fatty acids in 3-week-old animals. The retinal phospholipids in the 3-week-old guinea pigs had two main saturated fatty acids (16:0 and 18:0, accounting for 42–44% total fatty acids), one monounsaturated fatty acid (18:1n-9), three main n-6 PUFA (20:4n-6, 22:4n-6 and 22:5n-6) and one main n-3 PUFA (22:6n-3). The PUFA accounted for 34% to 36% of total fatty acids (Table 1). ALA was below the detection limit while the proportion of 18:2n-6 did not exceed 2%. Increasing the maternal dietary ALA from 0.7 to 17.3% total fatty acids was associated with two major changes in fatty acid composition: (a) a stepwise increase in DHA and (b) a stepwise decrease in 22:5n-6 (Figure 1). The increase in DHA was offset by a similar decrease in 22:5n-6, such that the total of DHA + 22:5n-6 in the retinal PL was very similar on each diet (21–22% of total fatty acids). DHA increased from 3.9% to 16.8% retinal PL fatty acids, while 22:5n-6 decreased from 16.9% to 5.6% as the dietary ALA proportions increased from 0.7% to 17.3%. On these five diets, the proportions of arachidonic acid (20:4n-6) were similar and ranged between 8.6% to 9.3% retinal PL fatty acids.

The brain and liver PL fatty acids from these animals showed similar trends to the retinal PL fatty acids in terms of the proportions of DHA and 22:5n-6. The fatty acid compositions of these two tissues are shown in Appendix A, Table A2 and Table A3, respectively. The proportions of DHA in the brain PL were lower than in the retina and ranged from 1.7% (diet LA:ALA of 103:1) to 6.9% (diet LA:ALA of 1:1). In the liver the proportions of long chain 20 and 22-carbon PUFA were less than 10% on all diets, with DHA values ranging from undetectable on the lowest ALA diet (LA:ALA of 103:1) to 0.91% (diet LA:ALA of 1:1). In both tissues, the proportions of 22:5n-6 changed inversely to those of DHA. AA proportions varied little across the diets in both brain (range: 9.4% to 10.5%) and liver (range: 4.9% to 5.9%).

## 4. Discussion

### 4.1. General Remarks about the Study

This study is the first to determine the amount of dietary ALA required to maintain retina PL DHA in the optimal retinal DHA range. To achieve this, pregnant guinea pigs were fed diets with increasing amounts of ALA at a constant LA level and their offspring were examined in early postnatal life (3-weeks of age). In the guinea pig, myelination commences in utero as it does in humans [15]. It was shown that at 3-weeks of age, ALA at 17.3% of diet fatty acids in a 1.1 ratio with LA achieved a retinal DHA value within the optimum range.

The present study explored using graded levels of dietary ALA to maximize retinal DHA levels. Other studies have compared the efficacy of dietary ALA versus DHA for sustaining retinal DHA levels, but no studies have examined the dose–response of dietary ALA on retinal DHA. Two of the studies comparing ALA with DHA have been in neonatal animals (pigs and primates); the third study was in growing guinea pigs from weaning to adulthood. In the piglet study, there were no differences in retinal DHA accumulation between two diets, which contained ALA at 4% diet fatty acids and the diet which contained DHA at 0.1% diet fatty acids [16]. In the baboon study, C13-labelled-ALA and C13-labelled-DHA were given orally and DHA was reported to be 12× more effective than ALA in being deposited as retinal DHA [17]. In the guinea pig study, it was found that DHA at 0.6% diet fatty acids resulted in similar retinal DHA values to when ALA was fed at 7.1% diet fatty acids [11]. In that study, the animals were fed from 3-weeks of age until 15-weeks of age (adults). The LA:ALA ratio in the diet was 2.3:1 and the retinal PL DHA value was 16.3% (within the optimal DHA range).

The novelty of this research is that it is the first to determine the amount of dietary ALA required to maintain retina PL DHA in the optimal retinal DHA range. It was found that a diet with LA:ALA in the range 1:1 optimized retinal DHA levels.

DHA is highly concentrated in neural membranes as well as in the photoreceptor cells of the retina and depletion of ALA in the diet leads to deficiencies of DHA in both tissues, as shown here. Since many studies have reported significant alterations in neural function associated with depletion of DHA in neural membranes as a result of the animals being fed ALA-deficient diets (7), the present work is also relevant to the brain.

### 4.2. What Is the Relevance of Retinal 22:5n-6?

Retinal and brain 22:5n-6 values relative to DHA are regarded as markers or indices to assess DHA status [18]. While DHA (22:6n-3) is the major PUFA in the retina and brain, these tissues can accumulate other 22-carbon PUFA (such as 22:5n-6) when diets are deficient in ALA [19]. In situations where dietary LA is present in excess of ALA and DHA is absent from the diet, the LA, which is the preferred substrate for the FADS2 enzyme, is free from competitive inhibition by ALA and is metabolized via the PUFA metabolic pathway to 22:5n-6 [20,21]. This PUFA is most likely synthesized in the liver [5] and transported to brain and retina tissues where it accumulates in these tissue PL. It is for this reason that tissue levels of 22:5n-6 are widely regarded as a biochemical indicator of a deficiency of ALA (and DHA) in the diet [7].

We showed that dietary ALA at 17.3% of diet fatty acids was able to maximize retinal DHA and minimize the biochemical marker of ALA-deficiency (22:5n-6). When the level of dietary ALA was between 1% and 6.4% of dietary fatty acids, retinal DHA values were significantly lower than 22:5n-6, indicating a state of biochemical ALA-deficiency.

### 4.3. How Is DHA Incorporated into the Brain and Retina?

An explanation as to why brain and retina are enriched in DHA could be related to DHA uptake by specific receptors in these tissues. In fact, recent work has shown a critical role for a transporter (Mfsd2a) expressed in the blood brain barrier and retina as a major transporter of DHA as 1-lyso-2-DHA-phosphatidylcholine into these tissues [22]. Other research has shown that the adiponectin receptor 1 is vital for the uptake of DHA by retinal pigment epithelial cells and that knock-out of this receptor decreases retinal DHA levels and is associated with significantly attenuated photoreceptor cell function [23]. It is presumed that DHA is supplied to the retina by different lipids in the plasma derived from the liver, including FFA, 1-lyso-2-DHA-phosphatidylcholine or triacylglycerol-fatty acids; however, this remains to be clarified [24]. Regardless of preferred lipid class for transport to the retina and brain, the liver is an important tissue for processing dietary fatty acids for further lipoprotein transport of esterified DHA for uptake by the retinal pigment epithelium and then transfer to the photoreceptor cells [25].

### 4.4. The Relevance of This Research Question to Humans

The relevance of this research question to humans is that throughout the world, ALA dietary intakes have been low, especially in relation to LA, which is the major PUFA in the food supply [26]. Typical LA:ALA values in human diets range from 15:1 (vegetarian and vegan diets [27]) to 8:1 in omnivore diets [28]. Furthermore, DHA in foods is essentially only available to those who choose to consume fish and other marine foods, who can afford fish or who can afford DHA supplements. Many populations exist on vegetarian diets, which are often devoid of sources of DHA and increasingly people in western countries are choosing to be vegans [27]. Additionally, droughts, famines, and war completely disrupt food supplies with no ability of people in these situations to choose food based on its nutrient content. However, one constant is that LA is found in abundance in most foods throughout the world [29]. Therefore, many people throughout the world have high LA intakes, low ALA intakes and little or no DHA.

The relevance of the present data to vegans is that the retinal PUFA showed evidence of significant biochemical n-3 (DHA) deficiency in 3-week-old guinea pigs with dietary LA:ALA of 2.75:1. In vegan and vegetarian diets, the LA:ALA is unlikely to be less than 10:1 [30,31]. Therefore, if these studies can be translated to vegans and vegetarians who do not consume sources of EPA and DHA, it suggests these groups do not have a sufficiently high ALA intake to optimize their retinal (and presumably brain) DHA levels. We further speculate that the only practical way for vegans and vegetarians to optimize their diet for optimal retinal DHA values is to consume suitable sources of DHA, as it is unlikely that they could reach a dietary LA:ALA ratio of 1:1.

### 4.5. Strengths and Weaknesses of the Study

This study used a dose–response approach to evaluate the ALA content of the diet required to achieve optimal retinal DHA levels in weaning guinea pigs. DHA levels in brain and liver were also assessed to support interpretation of the data. The weaknesses of the study include that only one time-point (3-weeks of age) was assessed; future studies should assess retinal DHA levels at several later time-points. Furthermore, such longer-term dose–response studies should evaluate retinal function.

## 5. Conclusions

The highest proportions of DHA in membrane lipids in mammals are found in the photoreceptor cells in the retina. Dietary deficiency of ALA can reduce retinal DHA by >80%, which leads to compromised retinal function. This study showed that increasing the maternal supply of ALA increased retinal DHA in a linear fashion in 3-week-old guinea pigs. The novelty of this research is that it is the first to determine the amount of dietary ALA required to maintain retina PL DHA in the optimal retinal DHA range. It was found that diets with LA:ALA in the range 1:1 optimized retinal DHA levels. We speculate that these studies may be relevant to vegans and vegetarians whose diets may not have sufficiently high ALA intakes to optimize their retinal DHA levels.

## Figures and Tables

**Figure 1 nutrients-14-00301-f001:**
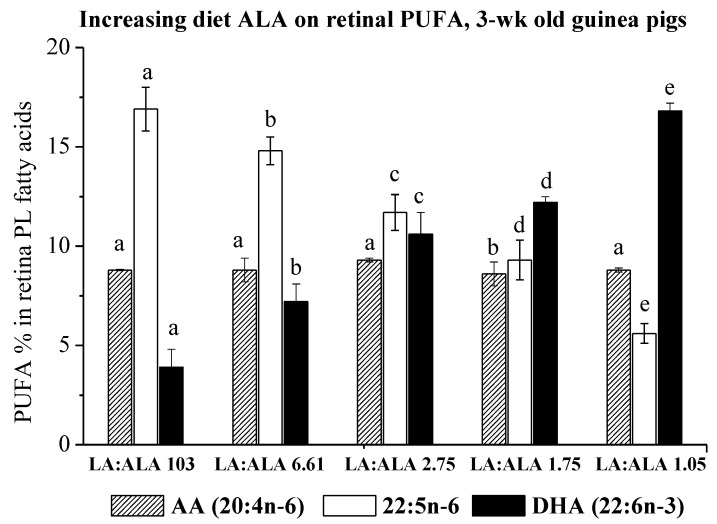
Effect of increasing dietary ALA on retina phospholipid proportions (mean ± SD) of selected PUFA (AA 20:4n-6), 22:5n-6 and DHA 22:6n-3) in 3-week-old guinea pigs whose mothers were maintained on these diets for the last one third of pregnancy. The number of animals in each group ranged from four to seven (see Table 1); the data with different letters for a particular fatty acid are significantly different (*p* < 0.05).

**Table 1 nutrients-14-00301-t001:** Effect of increasing dietary ALA on retinal fatty acid proportions (g/100 g phospholipid fatty acids) ^a^.

Diet Group ^b^	1	2	3	4	5
ALA% in diet lipid	0.7	2.8	6.4	10.0	17.3
Diet LA:ALA	103	6.61	2.75	1.75	1.05
(n) ^c^	(4)	(7)	(4)	(6)	(5)
Fatty Acids					
14:0	0.6 ± 0.0	0.6 ± 0.1	0.6 ± 0.0	0.6 ± 0.1	0.6 ± 0.0
16:0	20.3 ± 0.4 ^a^	19.1 ± 0.9 ^b^	19.5 ± 1.0 ^ab^	19.9 ± 0.8 ^ab^	19.1 ± 0.8 ^b^
17:0	0.5 ± 0.2	0.3 ± 0.0	0.3 ± 0.0	0.4 ± 0.0	0.3 ± 0.0
18:0	23.5 ± 0.0	22.6 ± 0.7	23.1 ± 0.3	23.7 ± 1.0	22.6 ± 0.3
16:1n-9	0.6 ± 0.0	0.7 ± 0.1	0.6 ± 0.1	0.6 ± 0.1	0.7 ± 0.1
18:1n-9	7.8 ± 1.2 ^a^	9.4 ± 0.7 ^b^	8.9 ± 0.8 ^b^	8.7 ± 0.8 ^b^	9.4 ± 0.3 ^b^
18:2n-6	1.9 ± 0.4 ^a^	1.7 ± 0.2 ^b^	1.6 ± 0.1 ^b^	1.6 ± 0.2 ^b^	1.7 ± 0.1 ^b^
20:3n-6	0.9 ± 0.1	0.8 ± 0.1	0.8 ± 0.0	0.8 ± 0.1	0.8 ± 0.0
20:4n-6	8.8 ± 0.0 ^a^	8.8 ± 0.6 ^a^	9.3 ± 0.1 ^a^	8.6 ± 0.6 ^b^	8.8 ± 0.1 ^a^
22:4n-6	4.2 ± 0.2 ^a^	3.7 ± 0.1 ^b^	3.6 ± 0.2 ^b^	3.1 ± 0.1 ^c^	3.0 ± 0.1 ^c^
**22:5n-6**	**16.9 ± 1.1 ^a^**	**14.8 ± 0.7 ^b^**	**11.7 ± 0.9 ^c^**	**9.3 ± 1.0 ^d^**	**5.6 ± 0.5 ^e^**
24:4n-6	2.5 ± 0.2 ^a^	2.3 ± 0.2 ^a^	2.1 ± 0.2 ^b^	1.9 ± 0.2 ^b^	1.6 ± 0.1 ^c^
24:5n-6	0.2 ± 0.0	0.2 ± 0.0	0.2 ± 0.0	0.1 ± 0.0	0.1 ± 0.0
18:3n-3	nd	nd	nd	nd	nd
20:5n-3	0.2 ± 0.0 ^a^	0.2 ± 0.1 ^a^	0.1 ± 0.0 ^a^	0.2 ± 0.1 ^a^	0.1 ± 0.0 ^b^
22:5n-3	0.4 ± 0.1	0.6 ± 0.1	0.9 ± 0.2	1.3 ± 0.1	1.5 ± 0.2
**22:6n-3**	**3.9 ± 1.0 ^a^**	**7.2 ± 0.9 ^b^**	**10.6 ± 1.1 ^c^**	**12.2 ± 0.3 ^d^**	**16.8 ± 0.3 ^e^**
24:5n-3	nd	0.2 ± 0.0 ^a^	0.4 ± 0.0 ^a^	0.6 ± 0.1 ^b^	0.8 ± 0.1 ^c^

^a^ Results shown as mean ± SD. The data with different superscripts are significantly different (*p* < 0.05); nd = not detectable. ^b^ The proportions of linoleic acid (LA) and alpha-linolenic acid (ALA) in diet group 1: LA 72.6%, ALA 0.7%; diet group 2 LA 18.6%, ALA 2.8%; diet group 3 LA 17.2%, ALA 6.4%; diet group 4 LA 17.5%, ALA 10.0%; diet group 5 LA 18.2%, ALA 17.3%. See Table A1 for details of diet fatty acids. ^c^ (n) = number of animals. The major fatty acid changes (22:5n-6 and 22:6n-3) are shown in bold.

## Data Availability

The data are available in the thesis by Lavinia Abedin, The Effect of Dietary Polyunsaturated Fatty Acids on Tissue Fatty Acid Levels of Guinea Pigs, RMIT University Library, RMIT University, Melbourne, 3001, Victoria, Australia.

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
