# Peer review of "Dietary Alpha-Linolenic Acid Supports High Retinal DHA Levels"

_nutrients, 2022, doi:10.3390/nu14020301_

Round 1
Reviewer 1 Report
Journal Nutrients (ISSN 2072-6643)
Manuscript ID: Nutrients-1543849
Type: Article
Title: Dietary alpha-linolenic acid supports high retinal DHA levels
Authors: Andrew J. James Sinclair * , Xiao-Fei Guo , Lavinia Abedin
I thank the Editor and the Editorial board for providing me an opportunity to review the above manuscript. I have few comments to improve clarity of the manuscript. The above manuscript can be accepted after minor revision.
Reviewer’s comments
- Abstract, Line 16 “At 16 weeks, retinal DHA levels increased compared to 3 weeks” sentence is not clear and it needs to be edited.
- What is the importance and significance of DHA in the retina needs to be explained in the introduction?
- Kindly mention what is the gestation period of guinea pigs and what embryo day the semi-synthetic diet was started to fed? And how it can be compared with human trimesters?
- In study 1, In study 1, 30 female and 7 male guinea pigs were housed together, and once pregnancy was confirmed the females were divided into 6 groups. Is all 30 females were pregnant at the same time? Need to mention how the pregnancy was confirmed in guinea pigs? How many females per group? Kindly describe the methodology in detail as it lacks clarity in its current form.
- I strongly recommend to include a graphical illustration of methodology and study design as the study design in the text is not clear. Kindly restructure it.
- How ALA deficiency is confirmed?
- On diet-fed with high ALA for 16 w, why there are higher DHA levels in the retina and brain but not in the liver?
- What are the recommended food sources that vegan can consume to increase their ALA levels?
- Tables 4 and 5 can be merged together for better understanding and clarity
- Table 2 and 3, diet group 2 data is missing.
- Effect of dietary ALA or DHA on the proportion of PUFA and the difference in PUFA levels in brain and retina tissues can be related to the difference in PUFA transport via blood-brain and blood-retinal barrier ?.

Author Response
Reviewer #1 Stated: I thank the Editor and the Editorial board for providing me an opportunity to review the above manuscript. I have few comments to improve clarity of the manuscript. The above manuscript can be accepted after minor revision.
Author’s general comment:
We found the reviewer’s comments very helpful and accordingly have made substantial changes to the manuscript, as outlined briefly here:
- We agree with Reviewer #2 that we describe 3 studies and the objectives of each were not the same. Therefore, we have removed Study 1b and Study 2 from the revised manuscript.
- We have significantly reduced the Discussion on the relevance of this work to humans (Section 4.5).
Our detailed responses to the reviewer’s remarks are shown below.
Reviewer #1 comments
- Abstract, Line 16 “At 16 weeks, retinal DHA levels increased compared to 3 weeks” sentence is not clear and it needs to be edited.
- Response: The manuscript has been significantly revised and we have deleted all reference to 16-week data (based on comments from one reviewer), so that the revised version only considers the effect of diet on retinal DHA in 3-week-old guinea pigs.
- What is the importance and significance of DHA in the retina needs to be explained in the introduction?
- Response: Thank you for making this point. The manuscript has been revised by the including this sentence (in bold) on lines 30-34 starting with Wheeler et al…:
“Introduction Docosahexaenoic acid (DHA, 22:6n-3) is concentrated in the disk membranes of rod outer segments of photoreceptor cells in the retina [1]. The high proportions of DHA in these membranes allows efficient conformational changes to occur in rhodopsin (the light receptor) during phototransduction [2]. Wheeler et al [3] showed that the electrical response of the photoreceptor cell was a relative function of dietary alpha-linolenic acid (ALA) and linoleic acid (LA) content, with the greatest response being to diets containing ALA. These data showed a selective functional role for omega 3 PUFA, presumably because of the very high DHA content of retina membrane phospholipids (PL).”
- Kindly mention what is the gestation period of guinea pigs and what embryo day the semi-synthetic diet was started to fed? And how it can be compared with human trimesters?
- Response 3a: The gestation period of the guinea pig is 69-71 days (Morrison et al 2018). The semi-synthetic diet was started on embryo day 47. This is now mentioned in the Methods section. “Once pregnancy was confirmed by the presence of a vaginal plug, the females were maintained on the commercial chow diets for the first 2/3 of pregnancy (until embryo day 46) and then the females in the 5 groups were placed on one of five semi-synthetic diets after that time until 3 weeks after delivery. The offspring were housed with the dams in their respective diet groups until the offspring were 3-weeks of age. The gestation period in the guinea pig is typically 69-71 days [12].
- Response 3b (comparison to human trimesters): Thank you for this comment. While there are similarities between humans and guinea pigs in terms of brain development in utero, using the timing for peaks in brain growth velocity as a marker of development, guinea pigs can be classified as prenatal brain developers, while humans are considered perinatal brain developers (and other rodents as postnatal brain developers McIntosh et al. 1979). We have added the following sentence to the Introduction (lines 84-89) to highlight that guinea pigs are increasingly regarded as a good model for various aspects of human physiology. “While there are similarities between humans and guinea pigs in terms of brain development in utero [12], using the timing for peaks in brain growth velocity as a marker of development, guinea pigs can be classified as prenatal brain developers, while humans are considered perinatal brain developers (and other rodents as postnatal brain developers) [13]. Like humans, the rapid phase of myelination and synapse formation is initiated in the last half of pregnancy in guinea pigs [12].
We also included the following sentence in the Methods section, lines 101-103: “The gestation period in the guinea pig is typically 69-71 days [12]. This relatively long gestation and the maturity of the guinea pig pups at birth means that a significant proportion of brain development, including myelination, happens in utero [14].”
- In study 1, In study 1, 30 female and 7 male guinea pigs were housed together, and once pregnancy was confirmed the females were divided into 6 groups. Is all 30 females were pregnant at the same time? Need to mention how the pregnancy was confirmed in guinea pigs? How many females per group? Kindly describe the methodology in detail as it lacks clarity in its current form.
Response: Thank you for these comments and we recognise that the description was not sufficient. Because we have eliminated the 16-week data (Study 1b), the revised details are as follows (Methodology, lines 94-104): “In this study , 21 guinea pigs (16 female and 5 male) were randomly divided into 5 groups consisting of 3-4 females per male guinea pig per group (4 groups of 3 females and 1 male; 1 group of 4 females and 1 male) and fed commercial chow diets. Once pregnancy was confirmed by the presence of a vaginal plug, the females were maintained on the commercial chow diets for the first 2/3 of pregnancy (until embryo day 46) and then the females in the 5 groups were placed on one of five semi-synthetic diets after that time until 3 weeks after delivery. The offspring were housed with the dams in their respective diet groups until the offspring were 3-weeks of age. The gestation period in the guinea pig is typically 69-71 days [12]. This relatively long gestation and the maturity of the guinea pig pups at birth means that a significant proportion of brain development, including myelination, happens in utero [14].
- I strongly recommend to include a graphical illustration of methodology and study design as the study design in the text is not clear. Kindly restructure it.
- Response. Thank you for this comment. We agree however since there is only one study reported in the revised manuscript, we do not feel this is necessary.
- How ALA deficiency is confirmed?
- Thank you. We have added a sentence in the Introduction (lines 57-59) which addresses your comment. “In studies of ALA deficiency in animals, no overt symptoms of the deficiency have been described unlike essential fatty acid deficiency [7], where animals show reduced weight, scaly skin, dandruff and loss of fertility. “
- We defined the biochemical indicator of ALA deficiency as a high tissue level of 22:5n-6. This was described in the Introduction lines 48-49: “Therefore, tissue levels of 22:5n-6 are widely regarded as a biochemical indicator of a deficiency of ALA (and DHA) in the diet [7].”
- On diet-fed with high ALA for 16 w, why there are higher DHA levels in the retina and brain but not in the liver?
- Response: The data from the 16-week study have been removed from the revised manuscript.
- What are the recommended food sources that vegan can consume to increase their ALA levels?
- Response: vegetable oils rich in ALA include flaxseed oil, perilla oil, canola oil. This has been mentioned in the Introduction lines 61-62, “the control diets used ALA-rich oils such as canola oil, rapeseed oil or soybean oil or fish oil (containing DHA)”
- Tables 4 and 5 can be merged together for better understanding and clarity
- Response: These data in Tables A4 and A5 relate to Study 2 which has been removed from the revised manuscript.
- Table 2 and 3, diet group 2 data is missing.
- Response: These data relate to Study 1b which has been removed from the revised manuscript.
- Effect of dietary ALA or DHA on the proportion of PUFA and the difference in PUFA levels in brain and retina tissues can be related to the difference in PUFA transport via blood-brain and blood-retinal barrier ?.
- Response: This is a reasonable suggestion. We attempted to address the uptake of DHA into the retina in Discussion section 4.4, lines 224-227. We have modified the start of this section with the inclusion of this sentence
“4.4. How is DHA incorporated into the brain and retina? An explanation why brain and retina are enriched in DHA could be related to DHA uptake by specific receptors in these tissues. In fact, recent work has shown a critical role for a transporter (Mfsd2a) expressed in the blood brain barrier and retina as a major transporter of DHA as 1-lyso-2-DHA-phosphatidylcholine into these tissues [22]. Other research has shown that the adiponectin receptor 1 is vital for the uptake of DHA by retinal pigment epithelial cells and that knock-out of this receptor decreases retinal DHA levels and is associated with significantly attenuated photoreceptor cell function [23]. It is presumed that DHA is supplied to the retina by different lipids in the plasma derived from the liver including FFA, 1-lyso-2-DHA-phosphatidylcholine or triacylglycerol-fatty acids, however this remains to be clarified [24]. Whatever the preferred lipid class is for transport to the retina and brain, the liver is an important tissue for processing dietary fatty acids for further lipoprotein transport of esterified DHA for uptake by the retinal pigment epithelium and then transfer to the photoreceptor cells [25].”
Reviewer 2 Report
The authors attempt to find out the optimal dose of DHA in the retina using dietary ALA. In a first study they fed pregnant guinea pigs with ALA in a range of 2.8% to 17.03% of the fatty acids in the diet and with a constant percentage of 17% of LA. DHA levels were studied at 3 months of age in the offspring. The results showed that retinal DHA increased in a linear fashion with the maximum on the diet with LA: ALA of 1: 1. At 16 weeks, retinal DHA levels increased compared with 3 weeks. Second, a 4-week study was conducted with ALA-deficient adult guinea pigs who fed a diet high in ALA (7.5%) or high in DHA (7.2% total n-3 fatty acids). A 0.5-fold increase in retinal DHA is observed with ALA and 3.4-fold with DHA. As a conclusion, it indicates that feeding diets for a long-term (during pregnancy and postnatally to 16 weeks) with high ALA levels supported high retinal DHA. Finally, he indicates as a conclusion that the current intakes of ALA in human diets, especially in relation to LA intakes, are not sufficient to optimize DHA levels in the retina.
Abstract
The Abstract generates some confusion, first it is indicated that pregnant women are fed and DHA is studied at three weeks of life in the offspring, but then it is indicated that an increase is observed at 16 weeks compared to 3, but without indicate that it was also analyzed at 16 weeks of age. It is difficult to understand clearly, the experimental design of the study would have to be better clarified. In addition, in the conclusion it indicates that it feeds until week 16 of postnatal life, but this is not indicated when explaining the design. Finally, the conclusion that human intake is not sufficient to optimize DHA levels in the retina cannot be supported by the results obtained in guinea pigs, it could be considered as speculation, but not as a conclusion.
Introduction
Although the introduction clearly shows the importance of DHA in the retina and the deficiency studies, a justification for the two studies indicated in the present work, especially the second, is not clearly observed. In addition, it is indicated that the aim of this study was to assess the level of dietary ALA necessary to increase retinal DHA to the optimal retinal DHA level, but in the previous paragraph it is indicated that a diet containing ALA at 7.1% (LA: ALA = 2.7) reached the target range of retinal DHA level (retinal DHA reached 16.4%), a study also carried out by the authors. I think the objective of the first study should be justified more clearly in the background and especially the reason for conducting the second study.
Material and Methods
In the abstract it is indicated that the proportion of LA is 17%, but in the material and methods 18% and then in results that go from 17.5% to 18.6%.
Why diets 1 and 2?
Where is the supplementary table (S1)?
Why were only the animals on diets 1 and 2 kept until week 16 and not the rest?
Why is the retina only removed in the second study and not the first?
Results
It indicates that five diets are carried out with an increase in ALA proportions, but in the material and methods there are 4 and in the abstract 6.
There is talk of an increase in DHA in the retina in the offspring at three weeks of age, but the material and methods do not indicate extraction of the retina.
When describing the results of 16 weeks of life with 3 weeks of life, the data in liver and brain are not shown, why?
In general there are many different studies with multiple data and they are difficult to follow and understand, the tables should be unified in some way that the results can be followed more clearly. Perhaps the data in the figures could be joined to the data in the tables.
Because when describing the results of study 2 we talk about refeeding and that term has not been used before?
Discussion
The discussion begins by stating the following: “This study is the first to determine the amount of dietary ALA required to maintain retina PL DHA in the optimal retinal DHA range. To achieve this, guinea pigs were fed diets with increasing amounts of ALA at a constant LA level in early postnatal life and to adulthood. " However, that is not correct, it is true that in a study there has been an increase from pregnancy to 3 weeks of life, but not to adulthood. The results obtained in study 1 (up to 3 weeks old) are mixed with those of study 1B (up to 16 weeks old), as if they were the same study, but they are not, they are only so partial.
Why is the relevance of tissue 22: 5n-6 discussed? It is something that was not in the objectives of the study and that was also already demonstrated in the introduction.
Nor do I understand sections 4.4 and 4.5, as an introduction they would be fine, but not in the discussion, at best it could be speculated how the results obtained could be transferred to humans.
Conclusion
More than conclusions they are speculations.
General
I do not understand the objective of the two studies, that is, each study should go separately, since they do not share the same objective, but rather seek or pursue different objectives. It should be further clarified why these two studies have been combined in the same publication.
In addition, there are actually three studies that are carried out, a study in which it is intended to see the effect of diets with an increase in ALA; Another comparing 16 weeks of life versus 3 weeks of life with two diets (one with a diet high in ALA and the other with a diet low in ALA) and finally a third who tries to see if a diet high in ALA for a short period of time can replenish DHA levels compared to a DHA diet.
I consider that there are too many studies or sub-studies that although they seem to be related, they are not clearly related or at least the authors do not connect them clearly at work, that makes reading the manuscript difficult to follow and it is as if they were reading three different studies at the same time.
Author Response
Reviewer #2
Author’s general comment:
We found the reviewer’s comments very helpful and have made a substantial changes to the manuscript, as outlined briefly here:
- We agree with Reviewer #2 that we describe 3 studies and the objectives of each were not the same. Accordingly, we have removed Study 1b and Study 2 from the revised manuscript.
- We have significantly reduced the Discussion on the relevance of this work to humans (Section 4.5).
Our detailed responses to the reviewer #2 remarks are shown below.
Reviewer #2 comments
The authors attempt to find out the optimal dose of DHA in the retina using dietary ALA. In a first study they fed pregnant guinea pigs with ALA in a range of 2.8% to 17.03% of the fatty acids in the diet and with a constant percentage of 17% of LA. DHA levels were studied at 3 months of age in the offspring. The results showed that retinal DHA increased in a linear fashion with the maximum on the diet with LA: ALA of 1: 1. At 16 weeks, retinal DHA levels increased compared with 3 weeks. Second, a 4-week study was conducted with ALA-deficient adult guinea pigs who fed a diet high in ALA (7.5%) or high in DHA (7.2% total n-3 fatty acids). A 0.5-fold increase in retinal DHA is observed with ALA and 3.4-fold with DHA. As a conclusion, it indicates that feeding diets for a long-term (during pregnancy and postnatally to 16 weeks) with high ALA levels supported high retinal DHA. Finally, he indicates as a conclusion that the current intakes of ALA in human diets, especially in relation to LA intakes, are not sufficient to optimize DHA levels in the retina.
Abstract
The Abstract generates some confusion, first it is indicated that pregnant women are fed and DHA is studied at three weeks of life in the offspring, but then it is indicated that an increase is observed at 16 weeks compared to 3, but without indicate that it was also analyzed at 16 weeks of age. It is difficult to understand clearly, the experimental design of the study would have to be better clarified. In addition, in the conclusion it indicates that it feeds until week 16 of postnatal life, but this is not indicated when explaining the design. Finally, the conclusion that human intake is not sufficient to optimize DHA levels in the retina cannot be supported by the results obtained in guinea pigs, it could be considered as speculation, but not as a conclusion.
- Response: Thank you for your helpful comments. We agree that the Abstract is confusing. We have revised the Abstract based on extensive revision of the manuscript (deleted studies 1b and 2). We have also made it clear that reference to humans is a speculation.
Revised Abstract reads:
Abstract: The retina requires docosahexaenoic acid (DHA) for optimal function. Alpha-linolenic acid (ALA) and DHA are dietary sources of retinal DHA. This research investigated optimizing retinal DHA using dietary ALA. Previous research identified 19% DHA in retinal phospholipids was associated with optimal retinal function in guinea pigs. Pregnant guinea pigs were fed dietary ALA from 2.8% to 17.3% of diet fatty acids, at a constant level of linoleic acid (LA) of 18% for the last one third of gestation and retinal DHA levels were assessed in 3-week-old offspring maintained on same diets as their mothers. Retinal DHA increased in a linear fashion with the maximum on the diet with LA:ALA of 1:1. Feeding diets with LA:ALA of 1:1 during pregnancy and assessing retinal DHA in 3-week-old offspring was associated with optimized retinal DHA levels. We speculate that the current intakes of ALA in human diets, especially in relation to LA intakes, are inadequate to support high DHA levels in the retina.
Introduction
Although the introduction clearly shows the importance of DHA in the retina and the deficiency studies, a justification for the two studies indicated in the present work, especially the second, is not clearly observed. In addition, it is indicated that the aim of this study was to assess the level of dietary ALA necessary to increase retinal DHA to the optimal retinal DHA level, but in the previous paragraph it is indicated that a diet containing ALA at 7.1% (LA: ALA = 2.7) reached the target range of retinal DHA level (retinal DHA reached 16.4%), a study also carried out by the authors. I think the objective of the first study should be justified more clearly in the background and especially the reason for conducting the second study.
- Response: Thank you for your helpful comments. We agree that the justification of the 16-week data and study 2 (refeeding) were not clear. We have deleted both these studies from the revised manuscript. The study has been justified as follows, Introduction, lines 71-83:
“What is not known is the minimum amount of dietary ALA to reach a retinal DHA value within the optimal retinal function range (19% DHA). We have shown in guinea pigs that ALA at 1% diet fatty acids (in a diet with LA:ALA of 17.3) was not sufficient to achieve the optimum retinal DHA levels (retinal DHA reached 9.6%), while a diet containing ALA at 7.1% (LA:ALA = 2.7) reached the target range of retinal DHA level (retinal DHA reached 16.4%) [11].
Therefore, the aim of this study was to conduct an ALA-dose response study to assess the level of dietary ALA necessary to increase retinal DHA to the optimal retinal DHA level and to minimize retinal 22:5n-6 in guinea pigs. To achieve this, pregnant guinea pigs were fed diets with increasing amounts of ALA at a constant LA level and the retina PL DHA and 22:5n-6 levels were measured in 3-week-old offspring. The hypothesis was that high ALA diets in a ratio with dietary LA of approx. 2:1 would lead to optimal retinal DHA levels.”
Material and Methods
In the abstract it is indicated that the proportion of LA is 17%, but in the material and methods 18% and then in results that go from 17.5% to 18.6%.
- Response: Thank you for your comments. These inconsistencies have been corrected.
Why diets 1 and 2?
- Response: Thank you for your comment. Diet 1 is the ALA deficient diet (see Methods, line 106, revised manuscript) which states “All diets contained 10% lipid (w/w); the 1st diet used safflower oil as its source of lipids and was used to provide a reference point for an ALA-deficient diet, and in the other 4 diets the lipids were supplied by a mixture of vegetable oils “……
Diet 2 has been removed from the revised manuscript.
Where is the supplementary table (S1)?
- Response: Thank you for your comment. This Table was mistakenly referred to as Supplementary Table. The data is actually found in Appendix Table A1.
Table A1. Fatty acid composition of the diets (g/100g fatty acids)
|
Diet Group |
1 a |
2b |
3b |
4b |
5b |
|
Fatty Acids |
|
|
|
|
|
|
12:0+14:0 |
nd |
10.9 |
23.7 |
13.8 |
10.9 |
|
16:0 |
8.2 |
28.0 |
18.9 |
24.0 |
13.7 |
|
18:0 |
2.9 |
4.0 |
3.5 |
4.1 |
4.3 |
|
18:1 |
15.3 |
34.5 |
29.8 |
30.0 |
35.3 |
|
18:2n-6 |
72.6 |
18.6 |
17.6 |
17.5 |
18.2 |
|
18:3n-3 |
0.7 |
2.8 |
6.4 |
10.0 |
17.3 |
|
20:1 |
nd |
nd |
nd |
nd |
nd |
|
20:4n-6 |
nd |
nd |
nd |
nd |
nd |
|
20:5n-3 |
nd |
nd |
nd |
nd |
nd |
|
22:5n-3 |
nd |
nd |
nd |
nd |
nd |
|
22:6n-3 |
nd |
nd |
nd |
nd |
nd |
|
18:2/18:3 |
103 |
6.6 |
2.75 |
1.75 |
1.05 |
aDiet 1 lipids consisted of safflower oil.
bDiets 3-6 consisted of mixed vegetable oils
(canola, coconut, palm stearine, safflower, sunola and flaxseed oils)
in order to have equal 18:2n-6 proportions in each diet.
Why were only the animals on diets 1 and 2 kept until week 16 and not the rest?
- Response: Thank you for your comment. The 16-week data which referred to animals on diets 1 and 2 has been removed from the revised manuscript
Why is the retina only removed in the second study and not the first?
- Response: Thank you for your comment. The removal of retinas in Study was reported in the Methods, and has now been highlighted in lines 112-116 “Three weeks after birth, offspring were sacrificed using carbon dioxide asphyxiation and the livers and brains were removed, washed in saline, blotted dry and at -70 oC for later fatty acid analysis. The retinas were quickly removed, washed in ice-cold phosphate-buffered saline and stored in chloroform/methanol (2:1v/v containing 10mg/L of BHT), at -70 oC for later fatty acid analysis..”
Results
It indicates that five diets are carried out with an increase in ALA proportions, but in the material and methods there are 4 and in the abstract 6.
- Response: Thank you for your comment. This has been revised and clarified with the removal of the 16-week study. Thus, the revised manuscript now refers only to five diets, lines 98-100. “and then the females in the 5 groups were placed on one of five semi-synthetic diets after that time until 3 weeks after delivery.”
There is talk of an increase in DHA in the retina in the offspring at three weeks of age, but the material and methods do not indicate extraction of the retina.
- Response: Thank you for your comment. The removal of retinas in Study was reported in the Methods, and has now been highlighted in lines 112-116 “Three weeks after birth, offspring were sacrificed using carbon dioxide asphyxiation and the livers and brains were removed, washed in saline, blotted dry and at -70 oC for later fatty acid analysis. The retinas were quickly removed, washed in ice-cold phosphate-buffered saline and stored in chloroform/methanol (2:1v/v containing 10mg/L of BHT), at -70 oC for later fatty acid analysis.”
When describing the results of 16 weeks of life with 3 weeks of life, the data in liver and brain are not shown, why?
- Response: Thank you for your comment. The revised manuscript no longer refers to the 16-week study.
In general there are many different studies with multiple data and they are difficult to follow and understand, the tables should be unified in some way that the results can be followed more clearly. Perhaps the data in the figures could be joined to the data in the tables.
Response: Thank you for your comment – we agree. The revised manuscript only contains reference to one study (3-week data) and no longer refers to the 16-week study or the refeeding study (study #2).
Because when describing the results of study 2 we talk about refeeding and that term has not been used before?
- Response: Thank you for your comment – we agree. The revised manuscript only contains reference to one study (3-week data) and no longer refers to the refeeding study (study #2).
Discussion
The discussion begins by stating the following: “This study is the first to determine the amount of dietary ALA required to maintain retina PL DHA in the optimal retinal DHA range. To achieve this, guinea pigs were fed diets with increasing amounts of ALA at a constant LA level in early postnatal life and to adulthood. " However, that is not correct, it is true that in a study there has been an increase from pregnancy to 3 weeks of life, but not to adulthood. The results obtained in study 1 (up to 3 weeks old) are mixed with those of study 1B (up to 16 weeks old), as if they were the same study, but they are not, they are only so partial.
- Response: Thank you for your comment. We agree and have deleted the 16-week (study 1B) and the second study from the revised manuscript.
Why is the relevance of tissue 22: 5n-6 discussed? It is something that was not in the objectives of the study and that was also already demonstrated in the introduction.
- Response: Thank you for your comment. We believe it is important to emphasise that 22:5n-6 is a biochemical indicator of n-3 fatty acid deficiency. We have included reference to 22:5n-6 in the Aims of the study and modified the Discussion section on 22:5n-6 by reducing its length.
AIMS (revised)” Therefore, the aim of this study was to conduct an ALA-dose response study to assess the level of dietary ALA necessary to increase retinal DHA to the optimal retinal DHA level and to minimize retinal 22:5n-6 in guinea pigs. To achieve this, pregnant guinea pigs were fed diets with increasing amounts of ALA at a constant LA level and the retina PL DHA and 22:5n-6 levels were measured in 3-week-old offspring.”
DISCUSSION (revised)“
4.2. What is the relevance of retinal 22:5n-6?
Retinal and brain 22:5n-6 values relative to DHA are regarded as a marker or index to assess DHA status [18]. While DHA (22:6n-3) is the major PUFA in the retina and brain, in certain circumstances these tissues can accumulate other 22-carbon PUFA (such as 22:5n-6), when diets are deficient in ALA [19]. In situations where dietary LA is present in excess of ALA, and DHA is absent from the diet, the LA which is the preferred substrate for the FADS2 enzyme is free from competitive inhibition by ALA and is metabolized via the PUFA metabolic pathway to 22:5n-6 [20, 21]. This PUFA is most likely synthesized in the liver [5] and transported to brain and retina tissues where it accumulates these tissue PL. It is for this reason that tissue levels of 22:5n-6 are widely regarded as a biochemical indicator of a deficiency of ALA (and DHA) in the diet [7].
We showed that dietary ALA at 17.3% of diet fatty acids was able to maximize retinal DHA and minimize the biochemical marker of ALA-deficiency (22:5n-6). When the level of dietary ALA was between 1% and 6.4% of dietary fatty acids, retinal DHA values were significantly lower than 22:5n-6, indicating a state of biochemical ALA-deficiency.
Nor do I understand sections 4.4 and 4.5, as an introduction they would be fine, but not in the discussion, at best it could be speculated how the results obtained could be transferred to humans.
- Response: Thank you for your comment. We believe that Section 4.4 is important in understanding how DHA is taken up by the retina. We have modified this section to read:
“Section 4.4. How is DHA incorporated into the brain and retina?
An explanation why brain and retina are enriched in DHA could be related to DHA uptake by specific receptors in these tissues. In fact, recent work has shown a critical role for a transporter (Mfsd2a) expressed in the blood brain barrier and retina as a major transporter of DHA as 1-lyso-2-DHA-phosphatidylcholine into these tissues [22]. Other research has shown that the adiponectin receptor-1 is vital for the uptake of DHA by retinal pigment epithelial cells and that knock-out of this receptor decreases retinal DHA levels and is associated with significantly attenuated photoreceptor cell function [23]. It is presumed that DHA is supplied to the retina by different lipids in the plasma derived from the liver including FFA, 1-acyl-DHA-phosphatidylcholine or triacylglycerol-fatty acids, however this remains to be clarified [24]. Whatever the preferred lipid class is for transport to the retina and brain, the liver is an important tissue for processing dietary fatty acids for further lipoprotein transport of esterified DHA for uptake by the retinal pigment epithelium and then transfer to the photoreceptor cells [25].”
We agree that Section 4.5 is too long. We have significantly reduced the length of this section and it now reads”
Section 4.5. The relevance of this research question to humans
“The relevance of this research question to humans is that ALA dietary intakes are low, especially in relation to LA, which is the major PUFA in the food supply throughout the world [26]. Typical LA:ALA values in human diets range from 15:1 (vegetarian and vegan diets [27]) to 8:1 in omnivore diets [28]. Furthermore, DHA in foods is essentially only available to those who choose to consume fish and other marine foods, who can afford fish or who can afford DHA supplements. Many populations exist on vegetarian diets which are often devoid of sources of DHA and increasingly people in western countries are choosing to be vegans [27]. Additionally, droughts, famines, and war totally disrupt food supplies with no ability of people in these situations to choose food based on its nutrient content. However, one constant is that LA is found in abundance in most foods throughout the world [29]. Therefore, many people throughout the world have high LA intakes, low ALA intakes and little or no DHA.
The relevance of the present data to vegans is that the retinal PUFA showed evidence of significant biochemical n-3 (DHA) deficiency in 3-week-old guinea pigs with dietary LA:ALA of 2.75:1. In vegan and vegetarian diets, the LA:ALA is unlikely to be less than 10:1 [30, 31]. Therefore, if these studies can be translated to vegans and vegetarians who don’t consume sources of EPA and DHA, it suggests these groups do not have a sufficiently high ALA intake to optimize their retinal (and presumably brain) DHA levels. We further speculate, that the only practical way for vegans and vegetarians to optimize their diet for optimal retinal DHA is to consume suitable sources of DHA, as it is unlikely that they could reach a dietary LA:ALA ratio of 1:1.”
Conclusion
More than conclusions they are speculations.
- Response: Thank you for your comment – we agree and have changed the final sentence of the Conclusions.
“Conclusions
The highest proportions of DHA in membrane lipids in mammals are found in the photoreceptor cells in the retina. Dietary deficiency of ALA can reduce retinal DHA by >80% which leads to compromised retinal function. This study showed that increasing the maternal supply of ALA increased retinal DHA in a linear fashion in 3-week-old guinea pigs. The novelty of this research is that it is the first to determine the amount of dietary ALA required to maintain retina PL DHA in the optimal retinal DHA range. It was found that diets with LA:ALA in the range 1:1 optimized retinal DHA levels. We speculate that these studies may be relevant to vegans and vegetarians whose diets may not have a sufficiently high ALA intakes to optimize their retinal DHA levels”.
General
I do not understand the objective of the two studies, that is, each study should go separately, since they do not share the same objective, but rather seek or pursue different objectives. It should be further clarified why these two studies have been combined in the same publication.
- Response: Thank you for your comment. We agree and have deleted the second study from the revised manuscript.
In addition, there are actually three studies that are carried out, a study in which it is intended to see the effect of diets with an increase in ALA; Another comparing 16 weeks of life versus 3 weeks of life with two diets (one with a diet high in ALA and the other with a diet low in ALA) and finally a third who tries to see if a diet high in ALA for a short period of time can replenish DHA levels compared to a DHA diet.
- Response: Thank you for your comment. We agree and have deleted the 16-week and the second study from the revised manuscript.
I consider that there are too many studies or sub-studies that although they seem to be related, they are not clearly related or at least the authors do not connect them clearly at work, that makes reading the manuscript difficult to follow and it is as if they were reading three different studies at the same time.
- Response: Thank you for your comment. We agree and have deleted the 16-week and the second study from the revised manuscript.
This manuscript is a resubmission of an earlier submission. The following is a list of the peer review reports and author responses from that submission.
Round 1
Reviewer 1 Report
The authors intend to study three different dietary approaches to optimize the DHA content in the retina. The scenarios investigated are: Increasing dietary ALA to pregnant guinea pigs (from 2.8% to 17.3% of diet fatty acids) and measure DHA levels in the retina in the offspring at three weeks of age; Comparing low doses of ALA or DHA on retinal DHA in adult guinea pigs fed from weaning; Comparing the efficacy of ALA or DHA to restore retinal DHA levels after a period on an ALA-deficient diet. Among the results, in the first scenario, an increase in DHA levels in the retina in the offspring at three weeks of life is observed. In the second scenario, it was observed that the DHA + ALA diet increased the DHA content in the retina 1.8 times more than the ALA diet. In the third scenario, an increase in DHA in the retina was observed in the DHA + EPA diet 2.2 times greater than the ALA diet. As a final conclusion, it indicates that the levels of DHA in the retina for a correct visual function is unknown.
Although the article seems interesting, it is not clear what its true objective is, since, in the introduction, what stands out is the importance of DHA in retinal function and that supplementation with ALA and preferably with DHA, indicating the existence of studies of supplementation, especially in deficient animals, with ALA and DHA in neuronal and visual function and ends by stating that there are no studies exploring dietary approaches to optimize DHA levels in the retina. To achieve this optimization, it chooses three different dietary scenarios, but at no point does it indicate the reason for those scenarios and, as with those scenarios, it can be verified which are the optimal levels of DHA for a correct retinal function, since it only measures the content of DHA, but not the visual function. They also measure DHA levels in the liver to see if they are a good biomarker of DHA levels in the retina.
In short, it is still just another study that aims to see the effect of supplementing an animal with different amounts of fatty acids on DHA levels.
I think it should be approached in a different way, highlighting the reason for the different selected scenarios and how these could affect the levels of DHA in the retina in different ways.
Material y Métodos
Are 6 groups established and indicate five diets, the sixth group which diet consumed? The ALA percentages of the first four diets and the percentages in the fifth diet should be better indicated. It is not clear why the authors use these diets
In the second scenario they add ascorbic acid, why don't they explain the reason for this supplementation? I believe that since one of the objectives is to check the effect of different dietary scenarios, focusing on their lipid profile, the percentage of fatty acids in these scenarios should be specified in more detail.
Discussion
In the discussion many aspects that are not directly related to the results obtained are dealt with in detail and that makes it difficult to follow.
In conclusion, he considered that so many different dietary scenarios create confusion when it comes to seeing and understanding the results. In my opinion, it should be shown the reason for the chosen dietary scenarios, what differences each would contribute to the increase in DHA levels in the retina and discuss each scenario separately, discussing the reason for the results obtained and what it contributes again to what that is already known.
Perhaps in this way the objective pursued can be better understood
Minor comments
Pag 2, line 82: “then” it is repeated
There are problems with the figures and tables, the figures are not clear, and the footers of the tables are confusing.
Reviewer 2 Report
This well designed animal study demonstrates the effect of maternal and post-natal diet on retinal DHA content. It builds on many known relationships between dietary ALA and DHA and retinal content. The key findings are the disconnect between liver DHA content and retina and the confirmation that DHA increases retinal DHA better than ALA.
The study was well-designed and well executed. The data is clearly presented. It contributes to the literature in the following way: 1. liver DHA does not reflect retinal DHA ( this is novel) 2. DHA raises retinal DHA better than ALA ( this was already known) The results are somewhat expected and not earth shattering or super high impact but still worthwhile to support existing literature and the novel connection between liver and retina.The discussion is nicely laid out and easy to follow.
Author Response
Response Letter to Reviewer 2
Reviewer stated:
This well designed animal study demonstrates the effect of maternal and post-natal diet on retinal DHA content. It builds on many known relationships between dietary ALA and DHA and retinal content. The key findings are the disconnect between liver DHA content and retina and the confirmation that DHA increases retinal DHA better than ALA.
The study was well-designed and well executed. The data is clearly presented. It contributes to the literature in the following way: 1. liver DHA does not reflect retinal DHA ( this is novel) 2. DHA raises retinal DHA better than ALA ( this was already known) The results are somewhat expected and not earth shattering or super high impact but still worthwhile to support existing literature and the novel connection between liver and retina.
The discussion is nicely laid out and easy to follow.
We thank the Reviewer for their positive comments. We have revised the manuscript taking into account the comments made. The revised Abstract is shown below.
Dietary deficiency of alpha-linolenic acid (ALA) reduces retinal docosahexaenoic acid (DHA) by >80% and compromises retinal function. This research investigated dietary strategies to optimize retinal DHA levels in guinea pigs using three dietary approaches. 1. Pregnant guinea pigs were fed dietary ALA from 2.8% to 17.3% of diet fatty acids, at a constant level of linoleic acid (LA) of 17%. Maximum retinal DHA levels in 3-week-old offspring occurred with LA:ALA of 1:1 in diet. 2. Weanling guinea pigs were fed 0.6% DHA + 1% ALA (of diet fatty acids), 1% ALA or 7% ALA for 12 weeks. Retinal DHA was 1.8-times greater in the DHA + ALA and the high ALA diet compared with the low ALA diet. 3. ALA-deficient guinea pigs were fed diets with equal total n-3 content (7.5% diet fatty acids as ALA or 7.2% DHA + EPA) for 4-weeks. Retinal DHA was 2.2-times greater in the DHA + EPA diet versus the ALA diet. Liver DHA values ranging from 0.84% up to 12.1% did not predict the high retinal DHA values (16-18%). Therefore, liver DHA was not a reliable biomarker of retinal DHA status. The retinal DHA level required for optimal retinal function is unknown.